# A Stress Management and Health Coaching Intervention to Empower Office Employees to Better Control Daily Stressors and Adopt Healthy Routines

**DOI:** 10.3390/ijerph22040548

**Published:** 2025-04-02

**Authors:** Despoina Ziaka, Xanthi Tigani, Christina Kanaka-Gantenbein, Evangelos C. Alexopoulos

**Affiliations:** 1Postgraduate Course on “The Science of Stress and Health Promotion”, Medical School, National and Kapodistrian University of Athens, GR-11527 Athens, Greece; 2First Department of Pediatrics, Aghia Sophia Children’s Hospital, Medical School, National and Kapodistrian University of Athens, GR-11527 Athens, Greece; 3Ergomneia Medical PCC, Ellispontou 11, GR-15669 Papagos, Greece

**Keywords:** health promotion, occupational stress, wellbeing, empowerment, coaching, lifestyle medicine, routine, occupational health, randomized control trial

## Abstract

The present pilot randomized control study examined the effectiveness of an 8-week stress management and health coaching intervention on perceived stress, healthy routines, sleep quality, self-efficacy, self-esteem and happiness. A total of 38 office employees were randomly assigned to the intervention group (IG, n = 20) or the control group (CG, n = 18) and validated tools were used to assess outcomes. Statistically significant differences in the IG after the 8 weeks were observed in perceived stress (i.e., a decrease in PSS-14 score, *p* = 0.043), in “Daily Routine”, i.e., an increase in control over the consistent timing of meals and sleep (*p* = 0.001) and in “Social and Mental Balance”, i.e., an increase in inclination to socialize, balance leisure and personal time and adopt positive thinking or cognitive control over stressors (*p* = 0.003). These improvements were reflected in an increase in total healthy lifestyle and personal control score (HLPCQ, *p* = 0.048). Short time and stress management and coaching interventions at workplaces can empower employees to increase control over stressors and to take the first step in adopting healthy behaviors by recognizing bad habits. Furthermore, in building sustainable employment, an empowered employee would participate at an organizational level more actively. Our preliminary results strongly support the idea that primary health care professionals should be educated in health coaching and relaxation techniques.

## 1. Introduction

The World Health Organization (WHO) emphasizes the necessity for healthy conditions in the workplace and defines “healthy work” as work where not only harmful conditions are absent, but where there are also sufficient and adequate conditions to promote health [1]. Work-related stress arises when an individual’s resources are insufficient to meet demands and emerges as a major challenge to the health and wellbeing of workers and to business success and sustainability [2]. In general, stress involves a harmful long-term imbalance between the individual’s resources and environmental demands. Cardiovascular diseases, musculoskeletal disorders, insomnia, lack of concentration and other mental disorders are a few of its effects [3]. Due to the extensive impact of stress in the workplace, its prevention and management are of particular interest not only to academic scholars, but also to governmental and industrial stakeholders [4]. Furthermore, psychological wellbeing is the ability of a person to feel satisfied and perform effectively regardless of the negative emotions that are usually a part of life. It is influenced by our ability to cope with stress in daily life through positive attitudes and achieving life goals. Individuals with high levels of self-esteem exhibit high self-efficacy, effective stress management, mental resilience and elevated levels of subjective happiness. However, managerial practices often have unintended consequences for employee wellbeing due to their multidimensional nature [2,5]. For example, by enriching duties or assigning more responsibilities, psychological wellbeing will increase, but may cause physical strain or overload; job rotations provide variety, but may place higher demands; incentives and compensation may enhance satisfaction, but may harm employees’ relationships and introduce inequity into the organization; team-building practices often increase social wellbeing, but may decrease psychological wellbeing in employees who strongly prefer to work independently.

As outlined by WHO, the workplace provides an appropriate setting for promoting health activities, including the protection of physical and psychological wellbeing and job satisfaction [1]. The framework for occupational health and safety is expanding to ensure the health and safety of employees by addressing all types of risks in a preventive manner, including the management of psychosocial risks and mental health conditions [6].

Interventions aim to address the sources of stress within the workplace at an organizational or team level, focusing on organizational restructuring or development (e.g., long working hours, lack of control over workload, time pressures, job insecurity, harassment, lack of sustainable communication, lack of effective conflict resolution, etc.) [7]. Interventions also target at the individual level and involve physical, cognitive or multimodal techniques and focus on alleviating existing stress and symptoms through stress management techniques, counseling and assistance programs. Cognitive interventions refer to techniques based on mindfulness and reducing negative thoughts, while multimodal interventions incorporate both physical and cognitive techniques and offer the highest potential for stress management [8,9,10]. On one hand, interventions that focus solely on the organization level may not be as effective for enhancing wellbeing as individual-level approaches. On the other hand, the effectiveness of stress management training is limited when the stressors are systemic or structural (e.g., excessive workload); therefore, the coping abilities of individuals may be insufficient in the long term when role restructuring or job redesigns are required.

In general, health promotion interventions encompass science-based stress management techniques, as well as promote a healthy lifestyle with an emphasis on healthy behaviors including physical exercise, balance nutrition, improving sleep quality, etc. These interventions help individuals to address daily stressors and ultimately improve their overall health and wellbeing [9,10,11,12,13]. By helping individuals focus on nutrition, physical activity, stress management, quality sleep, mental health and living with a sense of purpose, the risk and incidence of chronic disease are dramatically reduced. Four key health risk behaviors are lack of exercise or physical activity, poor nutrition, tobacco use and excessive alcohol consumption, which cause much of the disease, pain and premature death associated with chronic conditions [14]. The motivation of employees through health coaching is helpful to facilitate behavioral changes in these key lifestyle areas, fields that traditional healthcare approaches are not fully effective at supporting [15,16,17].

Social support, in terms of social relationships in the workplace, plays a crucial role in acknowledging and supporting employees, thereby making work more enjoyable, rewarding efforts and acknowledging the challenges presented by the workplace. Social support in the workplace can come from both colleagues and supervisors and has been associated with lower levels of professional burnout [18,19].

Many well-designed randomized controlled interventions (RCTs) in stress management have been published in various settings but relatively few have been at workplaces [9,10,20]. It is critical to implement well-designed interventions at workplaces, which can contribute to the prevention and management of the effects of chronic stress on employees’ health and productivity. Our intervention focuses primarily on employees utilizing coping strategies to manage stress (Lazarus and Folkman’s transactional model) rather than changing workplace conditions. We did not address or target occupational stress sources (e.g., by employing estimates of Demand–Control–Support or Effort–Reward imbalance).

The purpose of this study was to explore the possible effects of an 8-week stress management and health coaching intervention (the independent variable) on various outcomes including Perceived Stress, General Self-Efficacy, Subjective Happiness, Healthy Lifestyle and Personal Control, Sleep Quality and Self-esteem (as dependent variables).

## 2. Methods

### 2.1. Participants and Process

The study constitutes a 2-month, non-blinded, pilot randomized controlled trial (RCT), examining two groups: the intervention group (IG), which was trained in specific methods of stress management and the enhancement of positive health behaviors within a framework of an 8-week health promotion program, and the control group (CG), which attended only the presentation of the first week on stress overview. The study was conducted at the company P. Petropoulos SA’s premises in Athens, Greece (https://petropoulos.com/en/ accessed on 14 January 2025) after obtaining written administrative approval. The study protocol was approved by the Ethics Committee of the Medical School of the National Kapodistrian University of Athens (896/26/04/24).

A formal invitation was sent to office employees (N = 126) to participate in the intervention with a description of study procedures. The only exclusion criteria were the current or recent participation in psychotherapy groups or other stress management therapy sessions and the use of any psychotropic medication. All office employees who expressed interest and willingness to join signed the informed consent form. After obtaining the written informed consent, participants were randomized into two groups, the intervention group (IG) and the control group (waiting list) (CG), using the random number generator (https://www.random.org/ accessed on 15 November 2023). A total of 38 employees were finally included, with 20 in the IG and 18 in the CG. The questionnaires of the study were distributed in hard copies during the initial meeting and were completed and returned by the subjects within a week’s time in a sealed envelope to the occupational health clinic of the company. All personal information and data were confidential and were accessible exclusively to the researcher and the scientific supervisor (i.e., the occupational physician: ECA).

### 2.2. Intervention Group

The intervention team (IG and researchers) held a meeting once a week in a designated place (auditorium) within the company, lasting 50–70 min over the course of 8 weeks. During these sessions, widely used stress management techniques/tools were presented, alongside informative sessions about enhancing the understanding of the physiology and pathophysiology of stress. The program also included guidance on nutrition, physical exercise and sleep, as well as sessions focused on team empowerment overall.

A brief description of the weekly intervention is given in the diagram (Figure 1) and a detailed one as follows:-In week 1, an informative lecture was held concerning stress as well as related concepts (stress system; chronic stress effects: symptoms and diseases; stress and anxiety). At the end of the session, the intervention questionnaires were distributed to all participants (IG and CG). The control group did not participate in any other meeting and remained on a waiting list.-In the 2nd week, a lecture and training on lifestyle medicine, health behaviors and how they are linked to stress and chronic diseases as well as the process of changing harmful health behaviors took place. The purpose of the second meeting was to encourage active participation from the members of the IG with an emphasis on sharing personal experiences. Additionally, the importance of maintaining a journal as a measure of self-monitoring was analyzed. A weekly diary was distributed to the members to record their behaviors, set their goals and track their progress, alongside exercises and tasks related to the meeting’s topics.-In week 3, a lecture and training session on keeping a healthy routine and on specific lifestyle recommendations such as nutrition, physical exercise and the value of good sleep was conducted. Participants were given the opportunity to try foods rich in antioxidant properties. A new weekly diary was distributed to the members to record their behaviors, their goals and their achievements and additional exercises related to the meeting’s theme were assigned.-In the 4th and 5th weeks, training focused on scientifically validated stress management techniques after highlighting their benefits by communicating relevant studies. The techniques included diaphragmatic breathing, progressive muscle relaxation and guided mental imagery. The relaxation techniques focus on the conscious and controlled release of muscular tension. The combination of two or more relaxation techniques has proven to be effective in reducing stress [9,10,12]. Guided imagery is a stress management technique that shifts the participant’s attention to an envisioned mental image of flavors, sounds, sights, scents and emotions. The positive outcomes of this exercise are numerous, including stress reduction and aiding those who experience sleep disorders. In addition, during these two weeks, the participants were trained in additional skills using cognitive restructuring principles, the technique of worry interruption and the technique of using a gratitude journal. Cognitive restructuring is a process used in the cognitive behavioral therapy (CBT) approach, aimed at modifying dysfunctional thoughts, opinions, ideas and perceptions—referred to as cognitions—as well as behaviors that hinder an individual’s life [21]. The strategy of stopping thought involves deliberately following worry with a simple aversive consequence. The gratitude journal introduces the brain to the habit of scanning the surrounding environment for positive things [22]. New journals were distributed each week to allow participants to track the frequency of practicing with these techniques as well as to record lifestyle changes (e.g., exercise, diet and sleep). Exercises related to the themes of these three sessions were also assigned.-In week 6, a session centered on the concepts of self-awareness and communication as key tools for fostering positive health behaviors was held. It was emphasized that through self-awareness and effective communication—both with oneself and others—individual empowerment is enhanced, leading to improved health and lower stress levels. Finally, exercises and assignments related to the lecture topics were given to complete before the next meeting.-In week 7, to complete the intervention sessions, training on the concept of empowerment in the context of setting boundaries was conducted and the profile of a person with a resilient personality was presented, with particular emphasis on the characteristic of experiencing change as a challenge. Finally, exercises and assignments related to the lecture topics were given to complete before the next meeting.-In the 8th and last week, the scientific supervisor (occupational physician) addressed any remaining questions about the intervention and the intervention questionnaires were redistributed for completion and were returned within 10 days.

Throughout the entire duration of the intervention, the group was able to communicate daily via a group on the “Viber” application, a platform that ensures easy access for all participants, as it is installed on everyone’s mobile device for immediate connection. Through this application, the group shared general experiences related to the intervention, session recordings were shared and the researcher guided, inspired and supported the members, resolved any questions as well as reminded them of any pending tasks. There was also the opportunity for the group to participate in some voluntary activities related to the intervention, which took place outside of the office on Saturdays or Sundays. These actions aimed to foster familiarity among the team members, as well as with the researcher, to receive positive feedback and to motivate them to maintain their newly acquired habits as well as to enhance team spirit and the sense of social support in the workplace. Specifically, the actions involved contact with nature, contact with a sport and, on completion of the intervention, rewarding the members’ effort in a recreational area. The company supported these actions financially. In particular, the intervention group carried out two activities. In the first activity, the group visited a shelter in a mountain of Attica (“Parnitha”) with a rented bus, where they enjoyed nature, the company of others and toured its hiking trails, and in the second activity, the group visited a restaurant to deliver a meal.

### 2.3. Control Group (Waiting List Group)

In the control group, the same initial and final measurements as in the intervention group were performed. Group members participated in the first and eighth week of the intervention, as described above. After the completion of the intervention for the first group, the same procedure was followed with the control one, however no further analysis took place because due to various unforeseen reasons, many participants missed more than 2 sessions and only 8 completed the program properly.

### 2.4. Tools and Measurements

The measurements included demographic variables such as age, gender, marital status, weight, height, smoking habits, education and work-related information. The validated Greek language versions of the questionnaires administered for completion included:

1. The “Healthy Lifestyle and Personal Control Questionnaire (HLPCQ) [23]”. The HLPCQ consists of 26 sentences in which the individual indicates on a four-point Likert-type scale how often they perform each of the health behaviors listed in each sentence. The questionnaire tracks the lifestyle pattern of individuals who have increased control over their health, indicating their degree of empowerment, which is the main goal of contemporary health promotion programs and includes 5 subscales: a. Dietary Healthy Choices, representing control over food quantity and quality; b. Dietary Harm Avoidance, representing control over “food temptations”, such as stress-eating, binge-eating, soft drink and fast-food consumption; c. Daily Routine, representing the individual’s control over the consistent timing of meals and sleep; d. Organized Physical Exercise, representing the tendency to follow scheduled organized physical exercise and e. Social and Mental Balance, representing the individual’s inclination to socialize, balance leisure and personal time and adopt positive thinking or cognitive control over stressors. All factors were significantly positively related to each other, which indicates that they collectively represent the degree of empowerment and self-efficacy that a person possesses. A high HLPCQ score adequately reflects the lifestyle pattern of a self-efficacious, empowered individual and is consistently associated with lower scores in the Perceived Stress Scale (PSS).

2. The “Perceived Stress Scale (PSS-14) [24,25]”. The Perceived Stress Scale (PSS) is a 14-item self-report instrument that measures the degree to which situations in a person’s life are rated as stressful. It scores the frequency of feelings and thoughts within the past month on a 5-point Likert-type scale (from 0 = never to 4 = very often). There are seven positive and seven negative items, and the total score is calculated by summing the score of each item after reversing all positive items (minimum total score = 0, maximum total score = 56). Higher scores indicate a higher level of perceived stress in the last month.

3. The “Self-esteem scale” (SES) [26,27]. The self-esteem scale is used to assess the level of self-esteem. The questionnaire consists of 10 questions which the individual is asked to answer based on a Likert-type scale (1 = Never, 2 = Almost Never, 3 = Sometimes, 4 = Quite Often, 5 = Very Often) which expresses the degree (frequency) to which each statement applies to him/her. Higher scores indicate a higher level of self-esteem.

4. The “Subjective Happiness Scale (SHS) [28,29]”. The Subjective Happiness Scale is used to measure an individual’s subjective happiness. This scale consists of four items, two of which ask individuals to characterize themselves, both in absolute terms and relative to their peers’ ratings, while the other two briefly describe happy and unhappy individuals and ask respondents to answer how much they identify with each characterization. Higher scores indicate a higher level of Subjective Happiness.

5. The “General Self-Efficacy Scale (GSES) [30,31]”. The General Self-Efficacy Scale measures the individual’s belief in their own ability to handle challenging situations. The questionnaire uses 10 questions and the person answers depending on which answer expresses him/her: “Not at all True”, “Slightly True”, “Quite True” or “Absolutely True”. Higher scores indicate a higher level of general self-efficacy.

6. The “PITTSBURG Sleep Quality Index (PSQI) [32,33]”. The questionnaire uses questions related to sleep habits during the last thirty (30) days only. Responses should be as accurate as possible for the most days and nights during the last thirty (30) days. Higher scores indicate a lower level of Sleep Quality.

The scores of all the abovementioned scales, i.e., the PSS-14, GSES, SHS, HLPCQ total and its subscales (Dietary Healthy Choices, Dietary Harm Avoidance, Daily Routine, Organized Physical Exercise and Social and Mental Balance), PSQI and SES were used as dependent variables.

### 2.5. Statistical Analysis

For the descriptive measures of the sample, absolute (N) and relative (%) frequencies were used for the qualitative variables, and for the quantitative variables, the median with the range (Min–Max) and the mean with a 95% confidence interval (95% C.I.) were used. For comparisons between the two groups of descriptive characteristics, and comparisons of both the baseline study scores and the sub-scales scores, Fisher and Mann–Whitney U tests were used for the qualitative and quantitative variables, respectively, due to a small sample size. For comparisons in the IG group at baseline and 8 weeks after the intervention, both baseline and sub-scales scores were compared using the paired *t*-test. To analyze longitudinal changes in outcome measures, mixed-effects models with a random intercept were used. This approach accounts for the nested structure of repeated measurements within individuals while estimating the overall effects of time and group. A random intercept-only model was selected to adjust for between-subject variability in baseline levels while ensuring model stability, given the small sample size (N = 38) and only two time points per participant. A random slope was not included, as it could lead to model overfitting and convergence issues due to the limited number of measurements per individual. The statistical significance level was set at 5% and statistical analysis was performed using the SAS software package (version 9.4).

## 3. Results

The demographic characteristics of the participants are presented in Table 1. A total of 38 participants subjects participated in the study [20 intervention group (IG), 18 control group (CG)], of which 21 (55.3%) were male and 17 (44.7%) were female. According to the body mass index, 13 (34.2%) subjects were categorized as normal, 21 (55.3%) as overweight, 3 (7.9%) as obese, and 1 (2.6%) as underweight. Most participants were married (68.4%), highly educated (73,6%), nonsmokers (68,4%) and reported suboptimal coverage of their needs (65.8%) (Table 1). Half of the participants (52.6%) had more than 20 years of experience (Table 1). A statistically significant difference between the two groups was found in age (mean IG vs. CG: 42.9 vs. 50.8 years, *p* = 0.0016) (Table 1).

As illustrated in Table 2, at the start of the study, there were no significant differences in the scores between the IG and the CG (*p* > 0.05 for all scores). After the 8-week intervention, the IG differed significantly from the CG in healthy lifestyle and personal control (HLPCQ) score, in perceived stress (PSS-14), in the Subjective Happiness Scale (SHS) and in self-efficacy scale scores (GSES) (Table 2). Specifically, the HLPCQ score increased after the intervention (mean IG vs. CG before: 63.1 vs. 60.6, *p* = 0.12; mean IG vs. CG after: 66.3 vs. 60.6, *p* = 0.021), the PSS-14 score decreased (mean IG vs. CG before: 25.4 vs. 26.3, *p* = 0.30; mean IG vs. CG after: 21.4 vs. 26.3, *p* = 0.001), and the SHS score increased (mean IG vs. CG before: 4.9 vs. 4.4, *p* = 0.071; mean IG vs. CG after: 5 vs. 4.4, *p* = 0.025), as did the GSES score (mean IG vs. CG before: 28.9 vs. 27.4, *p* = 0.28; mean IG vs. CG after: 29.9 vs. 27.4, *p* = 0.031).

In Table 3, the scores of the “Social and Mental Balance” subscale of HLPCQ differed significantly between the two groups, both before and after the intervention (mean IG vs. CG before: 12.4 vs. 10.3, *p* = 0.024; mean IG vs. CG after: 14 vs. 10.3, *p* < 0.001). Scores on the other subscales did not differ significantly.

Compared to the baseline scores, significant differences were found for IG after the intervention in healthy lifestyle and personal control (HLPCQ) and in perceived stress (PSS-14) (*p* = 0.048 and *p* = 0.043, respectively) (Table 4). In addition, statistically significant increases found in the “Daily routine” and the “Social and Mental Balance” subscales of HLPCQ after the intervention in IG (*p* = 0.001 and *p* = 0.003, respectively) (Table 4).

Table 5 presents the results from the mixed effects models. The only significant group effect (*p* > 0.05) that emerged on the baseline scores of the study was the increase in the “Social and Mental Balance” subscale score of HLPCQ (b = 2.897, *p* = 0.001). It is worth mentioning the increase in self-esteem and self-efficacy scale scores, although not at a statistically significant level.

Statistically significant temporal effects were confirmed for PSS-14 (b = 2.105, *p* = 0.045), for the “Daily Routine” (b = −1.105, *p* = 0.011) and the “Social and Mental Balance” HLPCQ subscales (b = −0.815, *p* = 0.004) and marginally for the total HLPCQ score (Table 5).

## 4. Discussion

We explored the possible benefits of an 8-week stress management and health coaching intervention focused on learning stress management techniques and adopting a lifestyle that promotes health-positive behaviors in a group of office employees. We found that the participants improved their daily routine in terms of socializing, on time management, on control over stressors and on positive thinking. A slight increase in self-esteem and self-efficacy was also monitored.

Given the significant amount of time that adults spend in their workplaces each day, workplaces provide an appropriate venue for interventions related to healthy habits. However, these interventions are not always effective in changing bad health habits. The World Health Organization states that “simply providing information to patients is unlikely to change behavior. Health care providers need to understand the psychological principles underlying education and understand that motivating patients requires more skill than conveying brief information to the patient” [1,15,16,17]. Because the need to manage stress is so central to successful behavior change, stress should also be considered a key target in lifestyle change, along with nutrition, physical activity, smoking and alcohol consumption management and sleep. Although most providers are fully aware of the scientific theory for promoting health behaviors, they are usually not trained in behavior change beyond providing education and recommendations [16,17,18].

There is a broad consensus that a combination of multi-component interventions (focusing on lifestyle management that includes stress management or mindfulness training, physical activity, diet and control of tobacco and alcohol consumption) and health coaching is more effective [34], compared, for example, with well-planned interventions focusing only on obesity or stress management [11]. On the other hand, coaching programs have shown promising results [35], although there is a need for randomized controlled studies.

In most, if not all health care systems, primary care providers, and specifically physicians, are not able to support lifestyle changes. These changes demand long term and evidence-based guidance, education, encouragement and feedback by health coaching in the key lifestyle areas. Occupational physicians and other primary care physicians, or other health professionals, could and should act as health coaches (lifestyle medicine), and this is critical in fighting chronic diseases [17,18,19,36,37]. A health coach should act as a role model for a healthy lifestyle and as such, they should live health consciously while conveying information and expertise in an easy-to-understand manner.

Our study was developed within a framework of guidance and support to improve employee wellbeing and facilitate the achievement of health-related goals. We use health coaching techniques to engage participants in their care and improve their health. Our intervention seems promising as it showed quantitatively, according to the statistical results, beneficial changes in the management of perceived stress, in the promotion of a healthy lifestyle and personal control and especially in time management and balanced mental and social routine. Qualitative data recorded at each meeting (not presented here in detail) showed the positive attitude of the participants, who reported not being fully aware of the recommendations on key lifestyle factors (diet, physical activity, sleep, etc.), who managed to increase fruit consumption, physical activity, to relax when practicing stress management techniques, to improve their social relations by embodying a more positive attitude and by expressing their dissatisfaction, to improve their self-awareness and set goals for their personal development. Similarly, participants’ attitudes were positive in previous studies implementing multidimensional interventions. Given that the intervention was multidimensional, participants were able to benefit from different aspects and tools, as reflected in the experiences shared. The weekly diary distributed to the members to record their behaviors, set their goals and track their progress, alongside exercises and tasks related to the meeting’s topics was of great value. This tool introduces the brain to the habit of scanning the surrounding environment for positive things. In a related study, individuals who kept daily gratitude lists were more optimistic and satisfied, had better quality sleep and experienced less pain. Writing about stressful experiences has been found to reduce health care utilization in healthy samples [22]. Moreover, following the intervention activities, the participants strongly expressed the desire to continue such initiatives. Along with this request, a few employees expressed concerns regarding psychosocial stressors of occupational origin, contributing to work-related stress. This led to an immediate response from senior management in collaboration with the occupational physician to address these issues.

The strengths of the current study lie in its quantitative approach, in its randomized controlled trial design and the use of validated tools. Furthermore, it combines different stress-relaxation management techniques with health promotion in a mentoring–supportive context. Another significant strength of the intervention was the inclusion of recreational activities (excursion in nature, contact with a sport and social contact in a recreational area) which have a rewarding and encouraging effect in the effort to motivate and adopt healthy habits and stress management behaviors. The continuous monitoring, guidance and communication throughout the intervention by researchers with specialized knowledge in the evolving field of health coaching, the (qualitative) feedback of participants, and the expertise of the research team in relative research support the characterization of the intervention as holistic.

The limitations of our research include mainly the possibility of reporting bias since the outcomes were based on self-reports, the small sample size and the short follow-up period. Although selection bias was minimized through the random allocation of participants to groups (randomization), the mean age of the intervention group was significantly smaller than in the control group which might have had an impact on the observed changes. However, the difference in age was far smaller and insignificant when median age was considered. In addition, although validated questionnaires were used, we are aware that biased estimates of self-assessed behavior are relatively common in intervention evaluations (response bias). During an intervention the participants gradually trust the researcher and social desirability (“to look good” and/or “to answer in a desirable way”) bias commonly changes and also might happen due to and increase in respondents’ awareness of being study subjects. We believe that in our setting, social desirability bias was minimal because participants’ trust was well established due to the long-standing cooperation with the occupational physician and the open communication in all phases. An additional limitation is that we did not measure work-related factors such as job demands, job control or workplace social support.

Interventions in workplaces are always challenging and much remains to be learned about the factors that influence employee participation in wellness programs and improve their actual health behaviors. Studies have shown that participation rates in wellness programs tend to be low and that generally, healthier and less stressed workers have higher participation rates than more stressed workers who present greater health risks. After all, wellbeing and happiness are complex and multidimensional concepts and changes demand a considerable amount of time. So, short-term modifications to one’s health or lifestyle behavior may not have an immediate impact on an individual’s overall perception of life or general wellbeing, as such behaviors may need to be sustained over much longer periods to affect the more fundamental nature of wellbeing and quality of life. We did not find an increase in the subjective happiness score, although some of its primary characteristics such as self-esteem, self-efficacy, sociability and more effective coping with life’s stressors showed a positive trend in our study [29]. In addition, most studies such as the current one measured outcome immediately after the intervention and did not assess outcomes in the medium or long term to ensure that the benefits were maintained after the intervention.

Based on the above, as many other studies have shown in various groups [9,10,12,13,14,20], there is sufficient evidence to suggest that future interventional research of high quality should focus on multidimensional strategies. It is essential to better understand what influences the effectiveness of these interventions in general and in targeted occupational groups. It is proposed that future multidimensional studies expand their findings to larger samples using a greater variety of tools and longer follow-up periods. Lengthier follow-up studies will be necessary to examine the long-term effects of similar interventions. Future research should examine both individual-level and organizational-level stress interventions.

## 5. Conclusions

Based on our preliminary results, we strongly support the idea that primary health care professionals should be educated in health coaching and relaxation techniques. Short time stress management and coaching interventions at workplaces can empower employees to increase their control over stressors and adopt healthy behaviors by recognizing bad habits. Furthermore, an empowered employee would participate at an organizational level more actively by building sustainable employment. Employers commonly sponsor wellness programs, which mostly do not use valid methodologies. Besides design and development shortages, these interventions commonly suffer from implementation issues. For this reason, it is essential to engage experts in the early stage of a wellness initiative in advising the organization to maximize its validity and impact.

## Figures and Tables

**Figure 1 ijerph-22-00548-f001:**
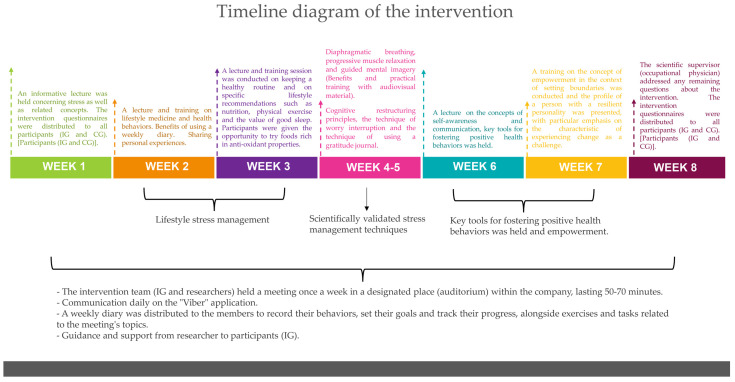
Timeline diagram of the intervention.

**Table 1 ijerph-22-00548-t001:** Demographic characteristics of intervention (IG) and control group (CG) participants.

	IG (n = 20)	CG (n = 18)	All (N = 38)	*p*-Value
Gender—n (%)				
Male	10 (50.0)	11 (61.1)	21 (55.3)	0.53
Female	10 (50.0)	7 (38.9)	17 (44.7)	
Body Mass Index (BMI)—n (%)				
Underweight	1 (5.0)	-	1 (2.6)	0.24
Normal weight	8 (40.0)	5 (27.8)	13 (34.2)	
Overweight	11 (55.0)	10 (55.6)	21 (55.3)	
Obese	-	3 (16.7)	3 (7.9)	
Marital Status—n (%)				
Single	6 (30.0)	1 (5.6)	7 (18.4)	0.13
Divorced	3 (15.0)	2 (11.1)	5 (13.2)	
Married	11 (55.0)	15 (83.3)	26 (68.4)	
Education—n (%)				
High School and/or Vocational Training	6 (30.0)	4 (22.3)	10 (26.3)	0.64
University/Technological Institute	9 (45.0)	8 (44.4)	17 (44.7)	
Postgraduate	5 (25.0)	6 (33.3)	11 (28.9)	
Coverage of needs—n (%)				
Barely or hardly	2 (10.0)	-	2 (5.2)	0.32
Moderately	14 (70.0)	11 (61.1)	25 (65.8)	
Fully	3 (15.0)	7 (38.9)	10 (26.3)	
Unknown	1 (5.0)	-	1 (2.6)	
Previous Experience—n (%)				
≤20 έτη	10 (50.0)	10 (55.6)	20 (52.6)	0.76
>20 έτη	10 (50.0)	8 (44.4)	18 (47.4)	
Smoker—n (%)				
Yes	3 (15.0)	2 (11.1)	5 (13.2)	>0.99
Former	4 (20.0)	3 (16.7)	7 (18.4)	
No	13 (65.0)	13 (72.2)	26 (68.4)	
Age (in years)				
Mean (95% CI)	42.9 (39.6–46.2)	50.8 (48.1–53.6)	46.7 (44.2–49.1)	**0.0016 ^§^**
Median (Min–Max)	46.0 (28–52)	51.0 (43–60)	46.5 (28–60)	
BMI (kg/m^2^)				
Mean (95% CI)	24.8 (23.5–26.2)	26.4 (24.8–28)	25.6 (24.6–26.6)	0.16 ^§^
Median (Min–Max)	25.1 (17.7–29.1)	25.9 (22.5–33.5)	25.5 (17.7–33.5)	

(^§^) Mann–Whitney U test *p*-value. All others are Fisher’s exact test *p*-values. In bold, *p* ≤ 0.05.

**Table 2 ijerph-22-00548-t002:** Scale scores in intervention and control groups before and after the intervention.

	Before the Intervention			After the Intervention		
Score	IG (n = 20)	CG (n = 18)	*p*-Value ^#^	IG (n = 20)	CG (n = 18)	*p*-Value ^#^
HLPCQ						
Mean (95% CI)	63.1 (57.1–69.1)	60.6 (56.6–64.5)	0.12	66.3 (60.3–72.3)	60.6 (56.6–64.5)	**0.021**
Median (Min–Max)	66.5 (33–81)	58.5 (50–78)		69.0 (36–86)	58.5 (50–78)	
PSS-14						
Mean (95% CI)	25.4 (22.4–28.4)	26.3 (24.4–28.3)	0.30	21.4 (17.5–25.3)	26.3 (24.4–28.3)	**0.001**
Median (Min–Max)	25.0 (12–39)	27.5 (16–32)		20.0 (10–50)	27.5 (16–32)	
SES						
Mean (95% CI)	22.3 (20.3–24.3)	20.2 (18.7–21.7)	0.054	22.5 (20.5–24.4)	20.2 (18.7–21.7)	0.069
Median (Min–Max)	22.5 (12–30)	19.5 (16–25)		23 (14–30)	19.5 (16–25)	
SHS						
Mean (95% CI)	4.9 (4.3–5.5)	4.4 (3.9–4.9)	0.071	5.0 (4.5–5.6)	4.4 (3.9–4.9)	**0.025**
Median (Min–Max)	5.3 (1.7–7.0)	4.3 (2.5–6.3)		5.3 (1.5–6.5)	4.3 (2.5–6.3)	
GSES						
Mean (95% CI)	28.9 (26.5–31.2)	27.4 (25.5–29.3)	0.28	29.9 (27.9–31.8)	27.4 (25.5–29.3)	**0.031**
Median (Min–Max)	29.0 (16–39)	27.5 (19–34)		30.0 (18–37)	27.5 (19–34)	
PSQI						
Mean (95% CI)	4.4 (3.3–5.4)	4.2 (3.2–5.1)	0.76	4.3 (3.1–5.5)	4.2 (3.2–5.1)	0.72
Median (Min–Max)	3.5 (2–11)	4.0 (0–9)		4 (1–13)	4 (0–9)	

HLPCQ: Healthy Lifestyle and Personal Control Questionnaire; PSS-14: Perceived Stress Scale; SES: Self-esteem scale; SHS: Subjective Happiness Scale; GSES: General Self-Efficacy Scale; PSQI: Pittsburgh Sleep Quality Index. (^#^) Mann–Whitney U test *p*-value. In bold, *p* ≤ 0.05.

**Table 3 ijerph-22-00548-t003:** HLPCQ subscales scores results before and after the intervention.

	Before the Intervention			After the Intervention		
Score	IG (n = 20)	CG (n = 18)	*p*-Value ^#^	IG (n = 20)	CG (n = 18)	*p*-Value ^#^
Dietary Healthy Choices						
Mean (95% CI)	15.7 (13.8–17.6)	15.2 (14.1–16.4)	0.65	15 (13.1–16.8)	15.2 (14.1–16.4)	0.69
Median (Min–Max)	15.5 (9–24)	15.0 (11–19)		15 (9–24)	15 (11–19)	
Dietary Harm Avoidance						
Mean (95% CI)	8.8 (7.8–9.7)	9.2 (8.0–10.3)	0.40	9.1 (8.0–10.1)	9.2 (8.0–10.3)	0.78
Median (Min–Max)	9.0 (4–12)	9.5 (4–12)		10.0 (4–13)	9.5 (4–12)	
Daily Routine						
Mean (95% CI)	21.6 (18.9–24.2)	21.8 (19.6–24.1)	0.86	23.7 (21.2–26.1)	21.8 (19.6–24.1)	0.15
Median (Min–Max)	22.5 (8–31)	23.0 (12–28)		24 (11–30)	23 (12–28)	
Organized Physical Exercise						
Mean (95% CI)	4.7 (3.7–5.7)	4.9 (4.0–5.9)	0.76	4.7 (3.7–5.7)	4.9 (4–5.9)	0.65
Median (Min–Max)	5.0 (2–8)	4.5 (2–8)		4 (2–8)	4.5 (2–8)	
Social and Mental Balance						
Mean (95% CI)	12.4 (11.0–13.8)	10.3 (9.2–11.3)	**0.024**	14 (12.6–15.3)	10.3 (9.2–11.3)	**<0.001**
Median (Min–Max)	12.5 (7–17)	10.0 (7–15)		15 (7–17)	10 (7–15)	

HLPCQ: Healthy Lifestyle and Personal Control Questionnaire; (^#^) Mann–Whitney U test *p*-value. In bold, *p* ≤ 0.05.

**Table 4 ijerph-22-00548-t004:** Differences in mean scores of scales and subscales scores (8 weeks–start of study) in the Intervention group.

Scale and Subscales	Mean Difference (95% C.I.)	*p*-Value ^#^
HLPCQ	−3.20 (−6.37, −0.02)	**0.048**
Dietary Healthy Choices	0.75 (−0.39, 1.89)	0.186
Dietary Harm Avoidance	−0.30 (−0.93, 0.33)	0.329
Daily Routine	−2.10 (−3.63, −0.57)	**0.001**
Organized Physical Exercise	0 (−0.92, 0.92)	>0.99
Social and Mental Balance	−1.55 (−2.48, 0.62)	**0.003**
PSS-14	4.00 (0.12, 7.88)	**0.043**
SES	−0.15 (−1.11, 0.81)	0.748
SHS	−0.15 (−0.58, 0.28)	0.470
GSES	−1.00 (−2.31, 0.31)	0.126
PSQI	0.05 (−1.22, 1.32)	0.935

HLPCQ: Healthy Lifestyle and Personal Control Questionnaire); PSS-14: Perceived Stress Scale; SES: Self-esteem scale; SHS: Subjective Happiness Scale; GSES: General Self-Efficacy Scale; PSQI: Pittsburgh Sleep Quality Index. (^#^) paired samples *t*-test *p*-value. In bold, *p* ≤ 0.05.

**Table 5 ijerph-22-00548-t005:** Results of random fixed effects mixed effects models with the baseline study scores (of scales and subscales) as response variables.

	b for Time	*p*-Value	b for Group	*p*-Value
HLPCQ total scale	−1.684	**0.051**	4.144	0.231
HLPCQ subscale				
Dietary Healthy Choices	0.394	0.183	0.102	0.921
Dietary Harm Avoidance	−0.157	0.323	−0.266	0.704
Daily Routine	−1.105	**0.011**	0.766	0.631
Organized Physical Exercise	<0.001	>0.999	−0.244	0.695
Social and Mental Balance	−0.815	**0.004**	2.897	**0.001**
PSS-14	2.105	**0.045**	−2.933	0.093
SES	−0.078	0.743	2.152	0.076
SHS	−0.079	0.463	0.573	0.103
GSES	−0.526	0.124	1.961	0.157
PSQI	0.026	0.933	0.158	0.807

HLPCQ: Healthy Lifestyle and Personal Control Questionnaire); PSS-14: Perceived Stress Scale; SES: Self-esteem scale; SHS: Subjective Happiness Scale; GSES: General Self-Efficacy Scale; PSQI: Pittsburgh Sleep Quality Index. Note: the reference category for the time is the ‘after 8 weeks’ scores and for the group the “control group” scores. In bold, *p* ≤ 0.05.

## Data Availability

The datasets presented in this study are available upon reasonable request from the corresponding author.

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
