# Peer review of "A Stress Management and Health Coaching Intervention to Empower Office Employees to Better Control Daily Stressors and Adopt Healthy Routines"

_ijerph, 2025, doi:10.3390/ijerph22040548_

Round 1

Reviewer 1 Report

Comments and Suggestions for Authors

Dear Author, 

You have done great work, I have minor suggestions for improving your manuscript.

Best regard,

Author Response

Dear Reviewer,

Thank you very much for taking the time to review our manuscript and for your kind feedback. Please find below our responses. The corresponding revisions are highlighted in text.

Comment 1: While the authors addressed some key methodological limitations, certain aspects could have been further emphasized to provide a more critical and transparent evaluation of the study. It is highlighted that the control group was significantly older than the intervention group, which may affect the interpretation of results. & Comment 2: The authors mention that the study had a relatively short duration and suggest that a long-term evaluation would be beneficial. The authors acknowledge that the small sample (N=38) limits the generalizability of the findings. However, it has not clearly stated whether participants were randomly assigned to groups, which could affect the validity of the findings.

Responses 1 & 2:  In Methods / Participants (Process) section (Section 2.1., page 3, lines 118-121) the randomization process was clearly stated: “After obtaining the written informed consent, participants were randomized into two groups, the intervention group (IG) and the control group (waiting list) (CG), using the random number generator (https://www.random.org/).”  The mean age was indeed significantly different due to the random allocation of a few younger employees in the intervention group. It can be seen in Table 1 however, that the difference in median age between groups was far smaller (and non-significant). In any case, we have acknowledged it (limitation section, page 12, lines 421-424):  “Although selection bias was minimized through the random allocation of participants to groups (randomization), the mean age of the intervention group was significantly smaller than in the control group which might have an impact on the observed changes. However, the difference in age was far smaller and statistically non-significant when median age was considered.”

Comment 3: In addition, although validated questionnaires were used, the authors do not discuss potential biases stemming from participants' subjective assessments. Therefore, my suggestion is that authors could add this limitation in the part of limitation of the study.

Response 3: Thank you very much for pointing this out! We have added text in limitation section (page 12, lines 419-420): “The limitations of our research include mainly the possibility of reporting bias since the outcomes were based on self-reports, the small sample size and the short follow-up.”  & (page 12, lines 425-434): “In addition, although validated questionnaires were used, we are aware that biased estimates of self-assessed behavior are relatively common in intervention evaluations (response bias). During an intervention the participants gradually trust the researcher and social desirability (“to look good” and/or “to answer in a desirable way”) bias commonly changes as also might happen due to increase respondents’ awareness of study subjects. We believe that in our setting social desirability bias was minimal because participants’ trust was well established due to the long-standing cooperation with the occupational physician and the open communication in all phases.”

Comment 4: The authors discuss their results in the context of existing literature, highlighting similarities and differences. But they primarily focus on significant results, while non-significant findings are briefly mentioned without further interpretation. They could discuss that.

Response 4: Thank you for the comment. Indeed, we have not discussed all the statistically insignificant results (e.g. sleep quality p>0,5). However, we have commented on happiness (discussion, page 12, lines 418-421): “We did not find an increase in subjective happiness score although some of its primary characteristics such as self-esteem, self-efficacy, sociability, more effective coping with life’s stressors showed positive trend in our study [29].” As far as we know, relative to our non-significant findings’ literature is poor and furthermore our study did not assess outcomes in the medium or long term. Our preliminary results mainly support the need for action towards primary health care providers training to support lifestyle change. We have further acknowledged that by adding text (page 13, lines 455-463): Lengthier follow-up studies will be necessary to examine the long-term effects of similar interventions. Future research should examine both individual-level and organizational-level stress interventions. Based on our preliminary results, we strongly support the idea that primary health care professionals should be educated in health coaching and relaxation techniques.

Reviewer 2 Report

Comments and Suggestions for Authors

The authors should be commended for their effort in conducting a randomized controlled trial (RCT) to evaluate a workplace stress management and health coaching intervention. The study addresses an important issue—workplace stress—and attempts to implement an evidence-based intervention aimed at improving employee well-being. The use of validated measurement tools strengthens the study’s methodological rigor.

However, given the small sample size (N=38), the study may be better positioned as a pilot study rather than a full-scale RCT. This should be explicitly acknowledged in the manuscript. Additionally, while the literature review provides a broad background, there is a need for a more explicit differentiation between transactional and interactional models of stress management, which would strengthen the theoretical foundation of the study.

Major Issues

  1. Theoretical Development and Conceptual Framework
    • The manuscript references both transactional and interactional models of stress but does not clearly distinguish between the two.
    • The intervention appears to be based on Lazarus and Folkman’s (1984) transactional model, which emphasizes individual coping mechanisms rather than structural changes to the work environment.
    • The Demand-Control-Support (DCS) model, on the other hand, is an interactional model where stress results from job demands, worker control, and social support. If this model is referenced, it should be explicitly clarified that the intervention does not alter work demands or worker control—it primarily focuses on modifying how employees respond to stress rather than changing workplace conditions.
    • The authors should revise the literature review to better situate their intervention within the transactional model rather than suggesting it aligns with an interactional approach.
  2. Control Group Data Post-Intervention
    • The study mentions that the control group later received the same intervention. However, there is no analysis of their data after completing the training. If these data were collected, the authors should incorporate them into the analysis. If not, the manuscript should acknowledge this as a limitation.
  3. Clarification of Dependent and Independent Variables
    • The manuscript does not explicitly define the independent and dependent variables in a clear manner.
    • The authors should specify:
      • Independent variable: Participation in the stress management and health coaching intervention.
      • Dependent variables: Scores on the Perceived Stress Scale (PSS-14), General Self-Efficacy Scale (GSES), Subjective Happiness Scale (SHS), and other relevant outcomes.
    • A clearer delineation of variables will improve the manuscript’s readability and methodological transparency.
  4. Statistical Methods and Explanation of Mixed-Effects Models
    • The manuscript utilizes univariate mixed-effects models, but there is no clear explanation of why this method was chosen or how it accounts for within-subject variability over time.
    • The authors should briefly describe the rationale for using mixed-effects models and explain the assumptions behind this statistical approach.

Minor Issues

  1. Abstract
    • The abstract is too long and should be condensed while maintaining key findings.
  2. Unclear Sentence on Managerial Practices (Lines 58-59)
    • The sentence regarding managerial practices and unintended consequences for employee well-being is vague. The authors should clarify the mechanisms through which managerial practices affect stress.
  3. Visualization of Intervention Timeline
    • The description of the intervention (Weeks 1-8) is difficult to follow in text format. A simple timeline diagram summarizing the weekly content would improve clarity.
  4. Table Formatting Issues
    • The authors should consult IJERPH formatting guidelines for table styles.
    • The asterisk (*) is commonly used to denote statistical significance (e.g.,p < .05), but in Table 2, it is unclear what it signifies. A footnote should be used instead.
    • Consider adding a vertical line in Table 2 to visually separate the intervention and control groups.
  5. Discussion on Psychosocial Work Factors
    • The discussion briefly mentions psychosocial stressors, but the study does not measure work-related factors such as job demands, job control, or workplace social support.
    • The authors should acknowledge this as a limitation and suggest future research that examines both individual-level and organizational-level stress interventions.
  6. Explanation of Mixed-Effects Model in Lay Terms
    • Many readers may not be familiar with univariate mixed-effects models.
    • The authors should include a brief, accessible explanation of why this model was chosen and how it enhances the analysis.

Conclusion

This manuscript presents a promising intervention for workplace stress management. The study's randomized design, use of validated measures, and focus on employee empowerment are commendable. However, the manuscript would benefit from stronger theoretical grounding, a clearer delineation of variables, and better presentation of control group data. Additionally, revisions to the abstract, tables, and graphical presentation would enhance readability.

With these improvements, the study will provide a more robust contribution to the literature on workplace stress interventions

Comments on the Quality of English Language

Mostly good, there are some awkward sentences.

Author Response

Dear Reviewer,

Thank you very much for your suggestions to improve our manuscript. Please find below our responses. The corresponding revisions are highlighted in text.

Comment 1: The study addresses an important issue—workplace stress—and attempts to implement an evidence-based intervention aimed at improving employee well-being. The use of validated measurement tools strengthens the study’s methodological rigor. However, given the small sample size (N=38), the study may be better positioned as a pilot study rather than a full-scale RCT. This should be explicitly acknowledged in the manuscript.

Response 1: We have acknowledged the specific comment in Abstract (lines 10 & 23), in Methods (first sentence, line 105) and in Conclusion sections (line 462, all highlighted in red).

Comment 2: Additionally, while the literature review provides a broad background, there is a need for a more explicit differentiation between transactional and interactional models of stress management, which would strengthen the theoretical foundation of the study. Theoretical Development and Conceptual Framework: The manuscript references both transactional and interactional models of stress but does not clearly distinguish between the two. The intervention appears to be based on Lazarus and Folkman’s (1984) transactional model, which emphasizes individual coping mechanisms rather than structural changes to the work environment. The Demand-Control-Support (DCS) model, on the other hand, is an interactional model where stress results from job demands, worker control, and social support. If this model is referenced, it should be explicitly clarified that the intervention does not alter work demands or worker control—it primarily focuses on modifying how employees respond to stress rather than changing workplace conditions. The authors should revise the literature review to better situate their intervention within the transactional model rather than suggesting it aligns with an interactional approach.

Response 2: Thanks for the comment. We have added text to differentiate between transactional and interactional approaches of stress management (Introduction, page 2, lines 68-73 & 95-98): Interventions that focus solely on the organization level may not be as effective for enhancing wellbeing as individual-level approaches. On the other hand, the effectiveness of stress management training is limited when the stressors are systemic or structural (e.g., excessive workload); therefore, the coping abilities of individuals may be insufficient in the long term when role re-structuring or job redesign are required. &

Our intervention focuses primarily on employees coping strategies to manage stress (Lazarus and Folkman’s transactional model) rather than changing workplace conditions. We did not address or target occupational stress sources (e.g. by employing estimates of Demand-Control-Support or Effort – Reward imbalance).

However, it is unrealistic to expect to remove all the potential stressors from an organization and job redesign may not be equally achievable across all professional contexts. Our intervention has elements of enhancing co-worker support and occupational risks specifically psychosocial ones were discussed under health coaching context. The possible effect of individual empowerment in organizational changes has pointed out (abstract, pre-last sentence): “Furthermore, an empowered employee would participate at an organizational level more actively in building sustainable employment”.

Comment 3: Control Group Data Post-Intervention: The study mentions that the control group later received the same intervention. However, there is no analysis of their data after completing the training. If these data were collected, the authors should incorporate them into the analysis. If not, the manuscript should acknowledge this as a limitation.

Response 3: The control group (n=18) received the same intervention with the only difference being that one action was implemented instead of two. However, due to various unforeseen reasons, many participants missed more than 2 sessions and only 8 completed the program properly. Thus, it was decided not to proceed with further analysis. We have added text (Section 2.3, page 5, lines 215-217): After the completion of the intervention for the first group, the same procedure was followed with the control one, however no further analysis took place because due to various unforeseen reasons, many participants missed more than 2 sessions and only 8 completed the program properly.

Comment 4: Clarification of Dependent and Independent Variables: The manuscript does not explicitly define the independent and dependent variables in a clear manner. The authors should specify: (a) Independent variable: Participation in the stress management and health coaching intervention. (b) Dependent variables: Scores on the Perceived Stress Scale (PSS-14), General Self-Efficacy Scale (GSES), Subjective Happiness Scale (SHS), and other relevant outcomes.

Response 4: We have clarified independent and dependent variables, by modifying relative text as requested (Introduction lines 99-102 and Section 2.4 lines 267-270).

The purpose of this study was to explore possible effects of an 8-week stress management and health coaching intervention (the independent variable) on various outcomes including Perceived Stress, General Self-Efficacy, Subjective Happiness, Healthy Lifestyle and Personal Control, Sleep Quality and Self-esteem (as dependent variables).

&

The scores of all the abovementioned scales i.e. the PSS-14, GSES, SHS, HLPCQ total and its subscales (Dietary Healthy Choices, Dietary Harm Avoidance, Daily Routine, Organized Physical Exercise and Social and Mental Balance), PSQI and SES were used as dependent variables.

Comment 5: Statistical Methods and Explanation of Mixed-Effects Models: The manuscript utilizes univariate mixed-effects models, but there is no clear explanation of why this method was chosen or how it accounts for within-subject variability over time. The authors should briefly describe the rationale for using mixed-effects models and explain the assumptions behind this statistical approach.

Response 5: We used mixed-effects models to account for the hierarchical structure of repeated measures data, modeling both fixed effects (time and group) and random effects (individual differences). A random intercept-only model was chosen to account for between-subject variability in baseline levels while ensuring model stability, given the small sample size and only two time points per participant. This model assumes that all individuals follow a common trajectory over time. The random intercept-only mixed-effects model accounts for between-subject variability by allowing each participant to have a unique baseline level (random intercept). However, it does not explicitly model within-subject variability over time, as a random slope was not included. Given that each participant had only two measurements, a random slope model was not feasible due to the risk of overfitting and convergence issues. Therefore, our approach balances model simplicity and stability while still capturing individual differences in baseline levels and estimating overall effects of time and group. Mixed-effects models also handle missing data more effectively than traditional repeated-measures ANOVA, making them a robust choice for longitudinal analysis."

Text is added in Statistical analysis section (page 6, lines 280-287): “To analyze longitudinal changes in outcome measures, mixed-effects models with a random intercept were used. This approach accounts for the nested structure of repeated measurements within individuals while estimating the overall effects of time and group. A random intercept-only model was selected to adjust for between-subject variability in baseline levels while ensuring model stability, given the small sample size (N = 38) and only two time points per participant. A random slope was not included, as it could lead to model overfitting and convergence issues due to the limited number of measurements per individual.”

Comment 6: Abstract: The abstract is too long and should be condensed while maintaining key findings.

Response 6: Done! Please see Abstract. 

Comment 7: Unclear Sentence on Managerial Practices (Lines 58-59): The sentence regarding managerial practices and unintended consequences for employee well-being is vague. The authors should clarify the mechanisms through which managerial practices affect stress.

Response 7: We added text to give examples on which and how managerial practices affect stress (Introduction, now lines 47-53): However, managerial practices often have unintended consequences for employee well-being due to their multidimensional nature [2, 5]. For example, by enriching duties or  more responsibility, psychological well-being increases but it may cause physical strain or overload; while job rotation provide variety it may place higher demands; incentive compensation may enhance satisfaction, but it may harm employees’ relationships and introduces inequity into the organization; team-building practices often increase social wellbeing but they may decrease psychological well-being in employees who strongly prefer to work independently.

.

Comment 8: Visualization of Intervention Timeline: The description of the intervention (Weeks 1-8) is difficult to follow in text format. A simple timeline diagram summarizing the weekly content would improve clarity.

Response 8: A time line diagram is added in Section 2.2.

Comment 9: Table Formatting Issues: The authors should consult IJERPH formatting guidelines for table styles. The asterisk (*) is commonly used to denote statistical significance (e.g.,p < .05), but in Table 2, it is unclear what it signifies. A footnote should be used instead. Consider adding a vertical line in Table 2 to visually separate the intervention and control groups.

Response 9: Done.

Comment 10: Discussion on Psychosocial Work Factors: The discussion briefly mentions psychosocial stressors, but the study does not measure work-related factors such as job demands, job control, or workplace social support. The authors should acknowledge this as a limitation and suggest future research that examines both individual-level and organizational-level stress interventions.

Response 10: Done (lines 432-434 and in lines 457-458)!  

An additional limitation is that we did not measure work-related factors such as job demands, job control, or workplace social support.

&

Lengthier follow-up studies will be necessary to examine the long-term effects of similar interventions. Future research should examine both individual-level and organizational-level stress interventions

Comment 11: Explanation of Mixed-Effects Model in Lay Terms: Many readers may not be familiar with univariate mixed-effects models. The authors should include a brief, accessible explanation of why this model was chosen and how it enhances the analysis.

Response 11: Mixed-effects model was chosen because it is well-suited for analyzing repeated measurements from the same individuals over time. This approach allows us to account for individual differences while examining overall trends in our data. Unlike traditional methods, mixed-effects models handle missing data more effectively and improve statistical power by considering all available measurements. Given our sample size, we used a random intercept-only model to balance model complexity with stability, ensuring a reliable analysis of group differences over time. In this study, we applied a random intercept-only model, which adjusts for between-subject variability by allowing each participant to have a distinct baseline level. This method is well-suited for longitudinal data as it efficiently handles missing values and provides a more flexible alternative to traditional repeated-measures ANOVA. Relative text is added (see 5th comment and response).